# Crack-Resistant Amino Resin Flame-Retardant Coatings Using Waterborne Polyurethane as a Co-Binder Resin

**DOI:** 10.3390/ma15124122

**Published:** 2022-06-09

**Authors:** Yanrong He, Yuzhang Wu, Wei Qu, Jingpeng Zhang

**Affiliations:** 1Research Institute of Forestry New Technology, Chinese Academy of Forestry, Xiangshan Road, Beijing 100091, China; 18600991247@163.com (Y.H.); wyz@caf.ac.cn (Y.W.); zhangjp@caf.ac.cn (J.Z.); 2Research Institute of Wood Industry, Chinese Academy of Forestry, Xiangshan Road, Beijing 100091, China

**Keywords:** melamine-urea-formaldehyde resin, intumescent flame retardant, crack-resistant, waterborne polyurethane, char formation

## Abstract

Surface cracking is a major issue in amino resin-based flame-retardant coatings, which can be reduced by mixing flexible resins into the coatings. In this study, flexible waterborne polyurethane (WPU) was added into a melamine-modified, urea-formaldehyde, resin-based intumescent flame retardant (MUF-IFR) coating. A molecular chain of WPU was inserted into the MUF network and formed a WPU/MUF-semi-IPN structure. The cracking resistance of the coating was gradually enhanced with the increase in WPU content. When the WPU content exceeded 25% of the total resin, there were no cracks in the coatings after crack-resistance tests. The coatings before and after toughening showed good transparency on wood surfaces. The influence of WPU on char formation and flame retardant properties were explored by TGA, SEM, and cone calorimetry. The results showed that the decomposition of WPU occurred before char formation, which decreased the integrity of the coating and damaged the compactness of the char. Therefore, the addition of WPU reduced the expansion height and the barrier capacity of the char as well as the flame retardant properties of the coating. When the amount of WPU was 25% of the total resin, compared to the non-WPU coating, the average heat release rate in 300 s (AveHRR_300s_) and the total heat release at 300 s (THR_300s_) of the samples were increased by 45.8% and 35.7%, respectively. However, compared to the naked wood, the peak heat release rate (pHRR_1_), AveHRR_300s_, and THR_300s_ of the samples with the coating containing 25% WPU were decreased by 64.2%, 39.0%, and 39.7%, respectively. Therefore, the thermal stability of WPU affected char formation. The amount of WPU added should be chosen to be the amount that was added just before the coating cracked.

## 1. Introduction

Transparent waterborne intumescent flame retardant (TWIFR) coatings can protect wood from fires without covering the unique texture of the wood [1,2]. Therefore, TWIFR coatings have received more and more attention [3]. The performance of such coatings greatly depends on their binder resins [4]. Amino resins are one of the most widely used and studied binder resins, and they show the best fire protection [5]. There are many advantages of amino resins, such as good rigidity, high-temperature resistance, and excellent bonding performance. They can also act as the gas source in intumescent flame retardant (IFR) systems [5,6]. However, amino resin coatings typically become hard and brittle after dried, which results in the surface cracking after a few months [7]. There are many amino (–NH_2_) and methylol (–CH_2_OH) groups in amino resins, and the curing of amino resins is achieved by the cross-linking between these groups [8]. However, there are some residual active sites that can be further cured after the film formation, resulting in the shrinkage of coatings and the generation of internal stress. When the internal stress exceeds the cohesion of the coating, it will crack [9]. In addition, the coating also easily cracks via substrate deformation because of its insufficient toughness. The high brittleness of a coating reduces its protective performance and degrades the surface quality of products [10]. Etherification is one of the main methods to reduce the number of active sites in amino resins [5]. However, most etherified amino resins are only soluble in organic solvents, which increases the VOC emission of coatings. In addition, excess alcohol decreases the water resistance of such coatings [11].

Polyvinyl alcohol (PVA) was used to form a semi-interpenetrating polymeric network (semi-IPN) with melamine-formaldehyde (MF) resins by Chen et al. [12]. A semi-IPN is a common toughening method for thermosetting resins [13,14,15] that is prepared by blending two polymers, one of which is a linear polymer (usually a thermoplastic) and the other is a network polymer (usually a thermoset). The flexible PVA chains bear stress, which improves the tensile strength of the system. When 12% PVA was added, the tensile strength of the MF resin increased by 18.64%. Polyurethane (PU) is a linear polymer composed of alternating soft and hard segments [16]. Due to their high toughness and impact energy absorption capacity, PU resins are often used as binder or co-binder resins [17,18]. They are introduced into epoxy (EP), polyacrylate, polysiloxane, vinyl ester resins, etc. to improve their flexibility by forming PU/EP-IPN [19,20,21], PU/polyacrylate-IPN [22,23], PU/polysiloxane-IPN [24], and PU/vinyl ester-IPN [25], respectively. PU was also added into MF rigid foams, and the resulting composite foams maintained the high compressive stress of MF and high tensile stress of PU [26,27]. Even though it has been widely used as a toughening resin, PU has not been applied in IFR coating systems. The influences of WPU on char formation and flame-retardant properties of coatings are still unclear.

In this work, a crack-resistant TWIFR coating was prepared based on WPU as the co-binder resin. The toughening and transparency of the coatings were characterized. The influences of WPU on char formation and flame-retardancy of the coating were explored by TGA, SEM, and cone calorimetry (CONE). After comprehensive evaluation, WPU showed an obvious toughening effect on the melamine-modified, urea-formaldehyde, resin/ammonium, polyphosphate/pentaerythritol, flame-retardant coating. Although the presence of WPU reduced the flame-retardancy of the coating, the coating still displayed a good flame-retardant performance compared with naked wood.

## 2. Materials and Methods

### 2.1. Materials

Formaldehyde solution (F, 37–40%), melamine (M), urea (U), and pentaerythritol (PER) were provided by Beijing Modern Eastern Fine Chemical Co., Ltd. (Beijing, China). Ammonium polyphosphate (APP, form II, n > 1000) was supplied by Sichuan Changfeng Chemical Co., Ltd. (Chengdu, China). Waterborne polyurethane resin (WPU, 25 °C, pH = 7–9, viscosity of 15′12″ (Tu-4 cup), solid content of 35%) was purchased from Guangdong Yuemei Chemical Co., Ltd. (Guangzhou, China). Sodium hydroxide (NaOH) and formic acid were supplied by Sinopharm Chemical Reagent Co., Ltd. (Shanghai, China). China fir (*Cunninghamia lanceolate Lamb.*) was purchased from Qingdao Huisheng Wood Industry Co., Ltd. (Qingdao, China).

### 2.2. Sample Preparation

#### 2.2.1. Synthesis of Melamine-Urea-Formaldehyde (MUF) Resin

The MUF resin was synthesized with a molar ratio of formaldehyde/melamine and urea (F/(M + U)) of 1.30 [28]. A total of 40 g urea and 120 g formaldehyde solution were added to a three-necked flask, and the pH was set to 8.0–9.5 with 5% NaOH solution. Subsequently, the mixture was heated to 90 °C for 30 min, and then the pH was adjusted to 5.0–5.5. In this period, the turbidity point method (which involved dropping the resin into 25 °C water) was used to judge the degree of condensation of the resin. Then, the remaining urea and melamine were added at pH = 7.5, and the mixture was kept for 30 min. Finally, the pH was adjusted to 8.5, and the MUF resin was obtained. The solid content and viscosity were 55% and 19 s (Tu-4 cup, 25 °C), respectively.

#### 2.2.2. Preparation of Waterborne Intumescent Flame-Retardant (WIFR) Coatings

The coating was prepared by mixing WPU resin and MUF resin as well as APP and PER physically. Moreover, the mixture was stirred for 5 min at 800 r/min. Based on the different proportions of WPU and MUF (0:100, 5:95, 10:90, 15:85, 20:80, 25:75, 30:70). The ratio of APP:PER was 3:1, which was 40% of the total resin. All ratios here were mass ratios. According to the amount of WPU, the samples were recorded as MIPU0-MIP30.

### 2.3. Characterization

#### 2.3.1. Transparency Test

Chinese fir was cut into wooden blocks with a smooth surface with dimensions of length × width × height = 100 mm × 100 mm × 20 mm. Then, the blocks were coated at a coating weight of 150 g/m^2^. After the coatings were dried completely at room temperature, their transparency was evaluated according to LY/T 3147-2019. The light transmittance of the coatings in the wavelength range of 400–800 nm was tested by a dual-beam UV-Vis spectrophotometer (TU-1901, Puxi, Beijing, China). The coatings were coated on a glass sheet (50 mm × 24 mm) at a coating weight of 150 g/m^2^.

#### 2.3.2. Cracking Resistance Test

The cracking resistance of the coatings was tested according to the standard method of GB/T 17657-2013 [29]. The coated wood specimens (100 mm × 100 mm × 20 mm) were heated in a drying oven at 70 ± 2 °C for 24 h and then placed at 25 °C and 50% relative humidity for 24 h. The surfaces of the specimens were observed with an electronic magnifying glass. The fracture surface of the coatings was characterized by field emission scanning electron microscopy (JSM-7610F, Hitachi, Tokyo, Japan) at an accelerating voltage of 8 kV after spray coating with a thin gold layer.

#### 2.3.3. Thermogravimetric Analysis (TGA)

The thermal stability of the coatings was tested by a thermogravimetric analyzer (DTA7300, Hitachi, Japan) from 25 °C to 800 °C. The heating rate was 10 °C/min with a gas flow rate of 50 mL/min under an air atmosphere.

#### 2.3.4. Cone Calorimetry (CONE) Test

The flammable properties of the coated specimens (100 mm × 100 mm × 20 mm) were characterized by a cone calorimeter (C3, Toyoseiki, Tokyo, Japan) in accordance with ISO5660-1 [25]. Before the tests, all samples were placed at 25 °C and 50% relative humidity for 24 h. During the tests, samples were wrapped in aluminum foil and exposed to a horizontal external heat flux of 50 kW/m^2^, with three replicates for each sample.

#### 2.3.5. Char Formation and Morphology

The char specimens of coatings were obtained by heating the cured coatings on steel (80 mm × 80 mm × 2 mm) at 800 °C in a muffle furnace (KSL-1100X-S, Hefeijingke, Hefei, China). The coating weight was 150 g/m^2^. The morphologies of the cross-section and the surface of char were characterized by field emission scanning electron microscopy (JSM-7610F, Hitachi, Japan) at an accelerating voltage of 8 kV after spray coating the samples with a thin gold layer.

## 3. Results and Discussion

### 3.1. Optical Transparency of the Coatings

The macro digital photos of the coated wood samples are shown in Figure 1. The texture of wood under the coating was clearly visible in all samples, and there were no differences in the transparency for all samples. According to the standard LY/T 3147-2019, these coatings were judged as “transparent”. To further explore the effect of WPU on the transparency of the coating, the light transmittance of coatings in the range of 400–800 nm was measured by a dual-beam UV-Vis spectrophotometer. As shown in Figure 2, the light transmittance of all samples was between 20–35%, which was inconsistent with results observed by the naked eye in Figure 1. When the substrate was wood with good permeability, the moisture of the coating could diffuse into the air and the interior wood during the drying process. Therefore, a uniform and dense coating with excellent transparency was formed. However, when the substrate was a glass sheet, moisture could only diffuse unidirectionally into the air. After the coating surface was dried, the moisture inside the coating was continuing to volatilize, resulting in a large number of microbubbles inside the coating. The existence of microbubbles led to a decrease in the light transmittance.

In addition, the transmittance increased slightly with the increase in WPU, which might be related to the viscosity of the coating. The viscosity and solid content of WPU were both lower than that of MUF. With the addition of WPU, the viscosity and solid content of the coating decreased, which facilitated the relaxing and rearrangement of flame retardant particles in the coating [2]. Without bubbles and flame retardant aggregation, light was scattered less when it passed through the coating [30].

### 3.2. Crack Resistance of the Coatings

The images of coatings after the crack resistance tests were taken by an electronic magnifier. As shown in Figure 3, there were dense and criss-cross cracks on the surface of the MIPU0 coating, indicating that the coating had poor cracking resistance when MUF was used as the polymer bonder alone. When WPU was introduced, the number of cracks decreased with the increase in the amount of WPU. When the amount of WPU exceeded 25% of the binder resin, there were no cracks in the coating. In order to investigate the reason, SEM was used to observe the fracture surface of the coating.

As seen in Figure 4, the fracture surface of MIPU0 was smooth and flat, without a trace of deformation, indicating that the coating did not yield and resisted fracture. Therefore, the fracture in MIPU0 was a brittle fracture [31,32]. After adding WPU, there were many filaments on the cross-section of the coating. Those filaments were caused by the yield deformation of the coating when it resisted fracture under an external force, which is characteristic of ductile fracture [33].Therefore, there was a state of coexistence of brittle and ductile fractures for the toughened coatings in the cross-section. WPU was interspersed in the network of MUF, forming a semi-IPN structure. The WPU molecular chain played a “crack nail anchor” role, which could disperse and transfer the forces acting on the MUF resin network, thereby reducing the fracture risk of the MUF resin network [34,35].

### 3.3. Thermal Stability of Coatings

The TGA and DTG curves of samples in an air atmosphere are shown in Figure 5a,b, respectively. The extracted data are given in Table 1. The decomposition of all samples could be divided into three stages, 30–200 °C, 200–350 °C, and 350–800 °C, which corresponded to the stable stage, the char formation stage, and the char oxidation stage, respectively. The weight loss of MIPU0 was 5.5% during the stable stage (30–200 °C), which was attributed to the release of bound water and the decomposition of small polymer molecules [36]. During the char formation stage (200–350 °C), APP broke down around 255 °C, releasing NH_3_ and H_2_O vapors and producing phosphoric acids (polymetaphosphate and pyrophosphate), which catalyzed the dehydration of PER. The molten resin formed the char skeleton. Moreover, MUF decomposed and released NH_3_ and CO_2_, blowing the char skeleton to form a swollen char at this stage [6,26]. The char formation stage was also a key stage for the intumescent flame retardant coating to play the role of flame retardancy. The final stage (350–800 °C) had a weight loss of 48.3%. During this stage, char was oxidized to CO_2_ by O_2_.

After WPU was introduced, a new weight loss rate peak occurred at 235 °C. According to the thermal degradation steps of WPU, this was attributed to the degradation of carbamate groups in the hard segment of the WPU [26]. This decomposition process produced isocyanates and polyols, which were further decomposed into amines, olefins, and carbon dioxide [37,38]. These gases escaped before charring and damaged the integrity of the coating [4]. In addition, the temperature of the maximum weight loss rate at 404.8 °C (*T_2max_*) and 634.4 °C (*T_3_*_max_) for MIPU0 moved towards around 380 °C and 580 °C for coatings with WPU, respectively. Compared with MIPU0, when the supplemental levels of WPU were 15%, 25%, and 30% of the total resin, the char residue at 800 °C decreased by 24.3%, 48.6%, and 41.9%, respectively.

### 3.4. Morphology of the Expanded Char

The flame retardancy of intumescent flame retardants is closely related to the strength and oxygen barrier capacity of the char [39,40]. The mechanical stability and thermal conductivity of char depend on the matching degree of the decomposition temperatures of the individual components of the flame retardant system [4,41].

Figure 6, Figure 7, Figure 8 and Figure 9 show the macrographs, the expansion heights, and the surface and internal SEM images of the char, respectively. The char surface of MIPU0 (Figure 6) was uniform and compact with good gloss. The greatest expansion height of MIPU0 was 11.6 mm (Figure 7). SEM images showed that the char surface of MIPU0 was continuous, dense, and fully expanded (Figure 8), and the internal pore size of the char was consistent and without damage (Figure 9). These indicated that MUF, APP, and PER cooperated well and formed an excellent swollen char during combustion [4,28]. When WPU was introduced into this system, the char surface was visibly discontinuous with poor gloss (Figure 6) and traces of shrinkage appeared, as well as holes (Figure 8). This implied that gases leaked during the char formation process. In the meantime, the internal pore sizes of the char (Figure 9) were inconsistent and there were some small holes in the pore walls. The interior pores transformed from the closed type (without WPU, Figure 1a) to the connected type (with WPU, Figure 1b). This was also the main reason for the decrease in the expansion height of the char (Figure 7). However, the height was decreased slowly upon increasing the WPU content. This indicated that the char height was sensitive to WPU but not its amount. The above changes in the morphology of the char before and after adding WPU indicated that the decomposition of WPU during the char formation stage damaged the compactness of the char, which was consistent with the results in Section 3.3.

### 3.5. Flame-Retardant Properties of Coatings

CONE tests can simulate fire conditions realistically [42]. The heat release rate (HRR) and the peak heat release rate (pHRR) are two important parameters used to determine the flammability of materials. The HRR and the total heat release (THR) curves of the coatings are shown in Figure 10a,b. The HRR and THR curves of the naked wood sample are shown in Figure 10c,d. The related data are listed in Table 2.

From Figure 10a, there were two obvious stages in the HRR curves of all the samples. The first stage was the formation of expanded char in the period 0–50 s. There was a heat release rate peak (pHRR_1_), which was attributed to the combustion of combustible gases generated by the decomposition of the coating. Compared with MIPU0, the pHRR_1_ of coatings with WPU were sharper, which was due to the rapid decomposition of WPU, indicating that the coatings with WPU burned more quickly and intensely [43]. After the formation of char, the flames were gradually extinguished by the isolation effect of the intumescent char. The second stage was in the period 50–300 s after the char broke. The char gradually failed under a continuous high temperature, and eventually the substrate was ignited. The time when the substrate was ignited was recorded as TTI_2_; this is not difficult to see from Figure 10a and Table 2. With the WPU content increased, TTI_2_ decreased. This was directly related to the quality of the char [44]. However, Figure 10a,b showed that the flame retardant performance of the samples with the coatings containing WPU did not decrease significantly with the increase in WPU content. When the amount of WPU was 25% of the total resin, compared to MIPU0, the TTI_2_ was reduced by 5 s, the average heat release rate in 300 s (AveHRR_300s_) and THR at 300 s (THR_300s_) increased by 45.8% and 35.7%, respectively. Nevertheless, compared to naked wood, as shown in Figure 10c,d, the pHRR_1_, AveHRR_300s_, and THR_300s_ of MIPU25 were decreased by 64.2%, 39.0% and 39.7%, respectively. This indicated that the flame retardant performance of the coatings with WPU were still good. From Table 2, there was no significant difference in total smoke production in 300 s (TSP_300s_), which indicated that the emission of the toxic smoke of WPU was suppressed by the flame retardant system.

## 4. Conclusions

In this work, a crack-resistant TWIFR coating was prepared by adding WPU as a co-binder resin. The coating was toughened via the formation of a semi-NPN structure between MUF and WPU resin. When the WPU content exceeded 25% of the total resin, there were no cracks in the coatings after crack-resistance tests. According to the LY/T 3147-2019 standard, the coatings before and after toughening showed good transparency on the wood surfaces. The addition of WPU improved the transparency of the coating. However, due to the low thermal stability of WPU, the thermal stability of the coating and the residual char were both reduced. The decomposition of WPU before the char formation damaged the compactness of the char layer and reduced the expansion height and barrier capacity of the char. When the amount of WPU was 25% of the total resin, compared to the non-WPU coating, the TTI_2_ was reduced by 5 s, and the AveHRR_300s_ and THR_300s_ increased by 45.8% and 35.7%, respectively. However, compared to the naked wood the pHRR_1_, AveHRR_300s_, and THR_300s_ were decreased by 64.2%, 39.0% and 39.7%, respectively. Therefore, the amount of WPU added should be chosen to be the amount that was added just before the coating cracked.

## Data Availability

The study did not report any data.

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
