# Peer review of "Crack-Resistant Amino Resin Flame-Retardant Coatings Using Waterborne Polyurethane as a Co-Binder Resin"

_materials, 2022, doi:10.3390/ma15124122_

Round 1

Reviewer 1 Report

The manuscript can be accepted for publication after some changes as suggested below:

1) there are grammatical errors, incomplete sentences, and typos throughout the manuscript, they should be corrected by a professional English speaker.

2) Error-values should be provided for all data presented in different tables.

3) It is not clear, how thick were the coatings, it should be clearly mentioned

4) long term durability of the coatings should be measured and included in the manuscript

Reviewer 2 Report

The authors of the study presented research on the influence of the flexible 

waterborne polyurethane addtion on the cracking of the flame-retardant coating. 

The research is focused mainly on microstructural and phisical properties of 

the coating. The research seems to be reasonably well performed, but some 

points need to be addressed.

All the text. Please check how's the Celsius degrees are written. In some 

places the degree sign should be changed to the upper index. e.g.  page 3 

subsection 2.3.5. "... on steel (80 mm × 80 mm × 2 mm) at 800 oC in a muffle 

furnace... "

In the case of the attached graphs, their size should be increased to increase 

the readability of especially Figures: 4, 5, 6, 8,9 and 10. 

In the case of the tested coatings, were also scratch tests performed? If not, 

it would be worth supplementing the submitted studies with such a study.

The results show that the use of WPU reduces the cracking tendency of the 

applied layers, and at the same time reduces the refractoriness of these 

coatings, with the emission of poisonous vapors during the test. Please comment 

on whether the amount of such vapors has been tested and whether the use of WPU 

will significantly affect the safety of people in the vicinity of elements 

covered with such coatings during the fire.

In the case of SEM images, characteristic areas should be indicated, which 

would definitely facilitate the interpretation of the results.

Reviewer 3 Report

The paper from He et al. investigates the use of a flexible waterborne PU in an intumescent FR formulation based on melamine-modified urea-formaldehyde resin, in order to increase the crack resistance of the latter. The paper is quite well written but the conclusions should be better supported by the experimental results, which are not impressive.

Some comments and suggestions are listed as follows:

  • it could be useful to perform TG analyses also in nitrogen, thus assessing the thermal stability of the IFR
  • Table 1: please- re-organize the table with Tonset, Tmax, residue at Tmax values and residues at the end of the test
  • Table 3: smoke parameters are completely missing
  • the overall English is acceptable, though some typo errors should be corrected

Reviewer 4 Report

Have a look my comments

Round 2

Reviewer 1 Report

The manuscript can be accepted after some english corrections.

Author Response

Thanks a lot for the reviewer's comments.

Reviewer 2 Report

The authors of the study presented research on the influence of the flexible waterborne polyurethane addtion on the cracking of the flame-retardant coating. The research is focused mainly on microstructural and phisical properties of the coating. The research seems to be reasonably well performed, but some points need to be addressed.

The authors took into account the proposed changes and the work may be published in its current form.

Author Response

Thanks a lot for the reviewer's comments. Some changes have been modified.

Reviewer 3 Report

The authors have revised the manuscript according to the Reviewer's comments and suggestions; therefore, the manuscript is now suitable for publication in its present form.

Author Response

(The authors gave the same response as above.)

Reviewer 4 Report

Hey,

You did say nothing about the recommended references either you have to add or write any objection

Author Response

Sorry for the absence of the comments about the references. The adviced references are about the PU as a binder to change the mechanical properties of the films, which are related to this topic. So the references have been added into this paper as No.17 and 18 references.